# Cervical Spine Instability Screening Tool Thai Version: Assessment of Convergent Validity and Rater Reliability

**DOI:** 10.3390/ijerph20176645

**Published:** 2023-08-25

**Authors:** Chanyawat Rueangsri, Rungthip Puntumetakul, Arisa Leungbootnak, Surachai Sae-Jung, Thiwaphon Chatprem

**Affiliations:** 1School of Physical Therapy, Faculty of Associated Medical Sciences, Khon Kaen University, Khon Kaen 40002, Thailand; janyawat.r@kkumail.com (C.R.); rungthiprt@gmail.com (R.P.); 2Research Center in Back, Neck, Other Joint Pain and Human Performance (BNOJPH), Khon Kaen University, Khon Kaen 40002, Thailand; 3Human Movement Sciences, School of Physical Therapy, Faculty of Associated Medical Sciences, Khon Kaen University, Khon Kaen 40002, Thailand; rung.smd28@gmail.com; 4Department of Orthopedics, Faculty of Medicine, Khon Kaen University, Khon Kaen 40002, Thailand; sursea@kku.ac.th

**Keywords:** cervical spine instability, screening tool, reliability, validity

## Abstract

Neck pain, dizziness, difficulty supporting the head for an extended period, and impaired movement are all symptoms of cervical spine instability, which may produce cervical spondylolisthesis in patients who have more severe symptoms. To avoid problems and consequences, early detection of cervical spine instability is required. A previous study created a Thai-language version of a cervical spine instability screening tool, named the CSI-TH, and evaluated its content validity. However, other characteristics of the CSI-TH still needed to be evaluated. The objective of the current study was to assess the rater reliability and convergent validity of the CSI-TH. A total of 160 participants with nonspecific chronic neck pain were included in the study. The Neck Disability Index Thai version (NDI-TH), the Visual Analog Scale Thai version (VAS-TH), and the Modified STarT Back Screening Tool Thai version (mSBST-TH) were used to evaluate the convergent validity of the CSI-TH. To determine inter- and intra-rater reliabilities, novice and experienced physical therapists were involved. The results showed that rater reliabilities were excellent: the intra-rater reliability was 0.992 (95% CI = 0.989 ± 0.994), and the inter-rater reliability was 0.987 (95% CI = 0.983 ± 0.991). The convergent validities of the VAS-TH, NDI-TH, and mSBST-TH when compared with the CSI-TH were 0.5446, 0.5545, and 0.5136, respectively (*p* < 0.01). The CSI-TH was developed for use by physical therapists and is reliable. It can be used by physical therapists, whether they are experienced or novices, and has an acceptable correlation to other neck-related questionnaires. The CSI-TH is concise, suitable for clinical use, and lower-priced when compared to the gold standard in diagnosis for patients with cervical spine instability.

## 1. Introduction

The cervical spine allows the widest range of motion relative to the other parts of the spine and supports the head’s weight. This leads to the cervical region being susceptible to a variety of disorders and complications [1]. Consequently, neck pain is considered one of the most common musculoskeletal problems and can impact individuals of all ages [2,3,4]. Previous epidemiological research found that the prevalence of neck pain ranged from 6.9% to 54.2% of the global population, with 34.8% to 48.5% having experienced neck pain at some point in their lives [5,6].

Cervical degeneration is a common cause of neck pain [7] and is generally divided into three stages, namely dysfunction, instability, and destabilization. Cervical spine instability occurs in the instability stage of the degeneration, which is predominantly observed in individuals aged between 35 and 70 years [8]. However, trauma, inflammatory arthritis, congenital collagenous compromise, and surgery involving the neck region can also cause cervical spine instability [9,10,11].

Cervical spine stability is critically important for three interrelated subsystems: the passive, active, and neural control subsystems [12]. In patients with cervical spine instability, the function of the cervical spine stability that maintains normal posture and protects structures such as vessels, nerves, and the spinal cord from the physiological load is compromised, resulting in the damage and irritation of these structures [13]. Additionally, pain or numbness in the arm, dizziness, inability to support the head for prolonged periods, hypermobility of movement, impaired movement, or cervical spondylolisthesis are symptoms related to cervical spine instability [14]. Furthermore, if the condition of cervical spine instability persists, a response occurs in the osseous and soft tissue adjacent to the affected vertebrae, which leads to nerve root and spinal cord involvement, resulting in a variety of signs and symptoms that may range from only neck pain to loss of muscle power and eventually a declining quality of life [15]. So, early detection of cervical spine instability is needed.

A common method to detect cervical spine instability is the X-ray assessment based on intervertebral kinematics during full cervical flexion and extension [14,16,17]. In X-ray films, when the cervical spine has a displacement of more than 3.5 mm or has an angle greater than 11 degrees, the patient is indicated to have cervical spine instability [18]. However, there are some limitations to the X-ray method, such as radiation exposure, specialist requirements, and ability to cope with situations with a large number of patients [19,20]. The questionnaire related to the common symptoms of patients with cervical spine instability may be useful for the screening process.

In 2005, Cook et al. reported the sixteen symptoms usually found in patients with cervical lumbar instability by using the Delphi study among specialists in musculoskeletal physical therapy [21]. These symptoms are also associated with other studies that reported the subjective examination of neck pain results in a diagnosis of cervical instability [14,22,23,24,25,26,27]. 

Rueangsri et al. (2022) decided to select the lists of subjective examinations reported by Cook et al. (2005) to establish a screening tool for Thai patients with cervical spine instability. They translated the sixteen items of the cervical spine instability screening tool from the English language into the Thai language and named it the Cervical Spine Instability Thai version (CSI-TH). Then, the CSI-TH was assessed for content validity by three specialist physiotherapists in musculoskeletal disorders and two orthopedic surgeons. They had at least 5 years of experience treating patients with neck pain, as well as being familiar with and understanding the signs and symptoms of cervical spine instability in patients. The assessment results show that the CSI-TH is a concise and intelligible questionnaire. Additionally, the content validity of the CSI-TH, as revealed by the index of item-objective congruence (IOC) value of 0.9 (95% CI = 0.82–0.98), demonstrates excellent content validity [28]. With this content validity value, it is reasonable to go further and assess other characteristics of CSI-TH screening. 

Existing questionnaires related to neck disorders are the Neck Disability Index (NDI), the Visual Analog Scale (VAS), and the Modified STarT Back Screening Tool (mSBST), which are the most common in clinical practice for assessment of pain intensity, personal care, and functional impairment [29]. These questionnaires have proven psychometric properties that ensure the usefulness of the tool in a clinical setting. The CSI-TH questionnaire relates to how the patient feels and his or her behavior, activity, and position adopted, some dimensions of which may correlate to the above-mentioned questionnaires.

The aim of the current study was to assess the inter- and intra-rater reliability as well as the convergent validity of the CSI-TH screening tool in comparison to the NDI, the VAS, and the mSBST. This evaluation was conducted among patients experiencing nonspecific chronic neck pain.

## 2. Materials and Methods

### 2.1. Ethics Statement

The current study was conducted between June 2022 and October 2022 after receiving approval from the Ethics Committee of the Center for Ethics in Human Research in Khon Kaen University in accordance with the Declaration of Helsinki (HE 642236). The participants were provided with information about the study and subsequently requested to sign an informed consent form after they had elected to participate in the study.

### 2.2. Study Population Recruitment

The participants could be either males or females, aged between 35 and 55 years. They had to have good communication and cooperation skills, as well as nonspecific chronic neck pain lasting more than three months. Neck pain is defined as discomfort in the posterior cervical spine, extending from the superior nuchal line to the spine of the scapula and from the superior border of the clavicle to the suprasternal notch, with or without referred pain up to (proximally to) the head or down the arm [30]. Patients who had tumors or metastases, or who were unable to communicate and read in Thai, were excluded from this study. 

Previous research by Sousa and Rojjanasrirat (2011) suggested that 10 participants for each question is reasonable for the number of participants in the reliability study. Since the cervical spine screening tool has 16 questions, the current investigation required 160 participants [31].

### 2.3. Procedure

Participants were recruited to the current study through an advertisement (i.e., a poster) and a face-to-face invitation. They were required to participate in two visits. On the first visit, after they had filled out a consent form, they were asked to provide information relating to (i) personal data (age, underlying disease, education level), (ii) neck pain symptoms, (iii) the Cervical Spine Instability Thai version (CSI-TH), (iv) the Neck Disability Index Thai version (NDI-TH), (v) the Visual Analog Scale Thai version (VAS-TH), and (vi) the Modified STarT Back Screening Tool Thai version (mSBST-TH). For the convergent validity evaluation, the NDI-TH, VAS-TH, and mSBST-TH were used to compare with the CSI-TH.

For the rater reliability testing, two researchers were involved. The first researcher (CR) was a novice orthopedic physical therapist with two years of clinical experience. The second researcher (AL) was an expert in physical therapy with six years of clinical experience. Participants were chosen at random, as regards whether they were interviewed by the novice or the expert for the first time. The researchers, CR and AL, were in separate rooms, so they were blinded to each other. The time taken for each interview was about 15 min. The researcher then made an appointment with the participants to collect further data. 

The second visit took place at least 48 h after the first day. This process was conducted by CR (the novice physical therapist). Participants were asked about their neck pain symptoms using the Global Rating of Change scale (GRC). Those participants who had scores between −3 and +3 on the GRC were included in the intra-rater reliability analysis, which assumed that they did not show any clinical change during the interval period [32]. Then, the second interview using the CSI-TH was completed. The study flow of the current study is shown in Figure 1. 

### 2.4. Questionnaires

#### 2.4.1. The Cervical Spine Instability Thai Version (CSI-TH)

The CSI questionnaire is made up of the 16 questions addressing clinical cervical spine instability symptoms that were discovered using the Delphi method from consensus among 172 qualified physical therapists who were orthopedic field experts in the United States [21]. The Thai version of the CSI questionnaire (named the cervical spine instability Thai version (CSI-TH))was translated from the English language to the Thai language by Rueangsri and co-workers in 2022 after they had had obtained authorization from the original cervical instability study via email [28]. It was designed for use by physiotherapists and demonstrated satisfactory content validity (0.6–1.0) [33]. The CSI-TH is shown in Table 1. Briefly, CSI-TH is designed to assess how the patient feels and their adopted posture of habit.

#### 2.4.2. Neck Disability Index Thai Version (NDI-TH)

The NDI is a self-administered questionnaire. The NDI was translated into the Thai language (NDI-TH), and its internal consistency was measured at an excellent level, with a Cronbach α value of 0.85 [34]. It comprises 10 sections designed to measure severity of pain, headaches, attention, and sleeping, as well as daily activities, including work, personal care, lifting, reading, driving, and recreation. The score for each section is from 0 to 5, with 0 expressing the highest level of function and 5 representing the lowest level of function. The maximum possible score is 50. The higher the overall NDI score, the greater the deficiencies. Most studies indicate that the NDI has adequate reliability, with intraclass correlation coefficients (ICCs) ranging from 0.50 to 0.98 [34,35,36]. For uncomplicated neck pain, the minimal detectable change (MDC) is around 5 points.

#### 2.4.3. Visual Analog Scale Thai Version (VAS-TH)

The VAS-TH is a simple measure that is often used to evaluate changes in pain intensity [37]. The VAS-TH is a 100 mm long straight line with no numbers on it. The far-left side is labeled “no pain,” while the far-right side is labeled “most severe pain”. The participant is instructed to cross or point down the line, and the researcher uses the ruler to measure the distance on the line that the patient has crossed [38]. Greater pain intensity is indicated by a higher VAS score. According to Boonstra et al. (2008) and Crossley et al. (2004), VAS has strong concurrent validity and reliability for measuring the severity of pain [39,40]. 

#### 2.4.4. The Modified STarT Back Screening Tool Thai Version (mSBST-TH)

The Modified STarT Back Screening Tool (mSBST) is a modified and translated (English to Dutch) version of the STarT Back Screening Tool (SBST) for back or neck pain patients [41,42] which is a quick screening tool that has the potential to help identify patient subgroups and direct the application of early prevention in primary care [43]. The mSBST consists of nine items, which can be divided into four items related to physical factors and five items related to psychosocial factors [41], and was translated into the Thai language in 2022 by Phungwattanakul and co-workers, becoming known as the mSBST-TH [44]. The mSBST-TH classifies patients into three groups based on their prognosis: low, medium, and high. Low-risk individuals are those with a total score of 3 or below. A medium-risk group is defined as those with a total score of 4 but a psychosocial sub-score of 3 or less. The high-risk group includes people with a psychosocial sub-score of 4 or higher [43]. The mSBST-TH has demonstrated high reliability (ICC = 0.81) for the total score and acceptable internal consistency (Cronbach α value = 0.73), with the MDC of the total score being calculated at 1.563 [44].

#### 2.4.5. The Global Rating of Change (GRC)

The global rating of change (GRC) is a scale that is designed to evaluate changes in a patient’s clinical symptoms over time [45]. The questions relate to the patient’s condition compared to the baseline. There are 15 points of GRC on a −7 to +7 scale; 0 is “no change”, −7 is “a very great deal worse”, and +7 is “a very great deal better” [46]. Regarding responsiveness, GRC has shown a moderate correlation between NDI change scores (r = 0.46, *p*-value < 0.001) [32]. For the inter-rater reliability between clinician- and patient-rated, the GRC reached an ICC value of 0.74 and achieved a Pearson’s correlation of 0.72–0.90 when compared with the patient-rated importance of change [45].

### 2.5. Statistical Analysis

To express the participants’ baseline characteristics, the percentages, means (standard deviations), and minimum and maximum values were used as a descriptive analysis. The STATA statistical package, version 14.0 (StataCorp, College Station, TX, USA), was utilized for reliability and validity considerations. The rater reliabilities were assessed by using the Intraclass Correlation Coefficient (ICC) model (3, 1) for the total score. The Neck Disability Index (NDI-TH), Visual Analog Scale (VAS-TH), and Modified STarT Back Screening Tool (mSBST-TH) were applied to evaluate the convergent validity of the CSI-TH and expressed using the Pearson correlation coefficient. The interpretation of correlation coefficients adhered to the guidelines outlined by Portney and Watkins (2009) which categorize correlation as follows: 0.00 to 0.25 = little or no relationship, 0.25 to 0.50 = fair relationship, 0.50 to 0.75 = moderate-to-good relationship, and above 0.75 = good-to-excellent relationship [47]. 

## 3. Results

The demographic characteristics of the 160 participants are shown in Table 2. The mean age of participants was 44.60 ± 6.51 years, with females accounting for 80 percent and males accounting for 20 percent. The mean duration of the pain was 15.48 ± 19.26 months.

### 3.1. The Score of Each Questionnaire

The score of each questionnaire, including the mean, median, minimum score, maximum scores, quartile 1 (Q1), and quartile 3 (Q3) is shown in Table 3.

### 3.2. Rater Reliability Consideration

The inter-rater reliability of this screening tool as between expert and novice physiotherapists revealed an excellent agreement value of 0.983 (95% CI = 0.983–0.991). In terms of intra-rater reliability, the novice physical therapist used the CSI-TH tool twice with a 48 h interval to minimize any recall of prior answers and alterations in clinical neck status. It showed excellent intra-rater reliability, with a high agreement value of 0.995 (95% CI = 0.993–0.998) (Table 4).

### 3.3. Convergent Validity Assessment

Table 5 shows the Pearson correlation coefficients as between the CSI-TH and the other questionnaires, namely the VAS-TH, NDI-TH, and mSBST-TH. The correlation between the CSI-TH and the VAS-TH, NDI-TH, and mSBST-TH was moderate [47]. A statistically significant correlation exists between the values of 0.5136 and 0.5545 (*p* < 0.01).

## 4. Discussion

Although the symptoms of patients with cervical instability have been previously documented [21] and translated into the Thai language [28], there is currently no available report on the validation (in comparison with the remaining questionnaires) and reliability of screening tests for these symptoms. Given that cervical instability constitutes a significant clinical concern for patients and has the potential to result in spinal cord irritation [21], there arises a clear necessity for early screening methods to effectively identify and address this condition. 

In a previous study conducted by Rueangsri et al. (2022), a list comprising 16 items associated with cervical spine instability was translated from the English language into the Thai language. This translated version was named the Cervical Spine Instability Thai version (CSI-TH). Their study further demonstrated the exceptional content validity of the CSI-TH [28]. Therefore, in order to enhance the substantiation of the capacities of the CSI-TH, the present study focused on examining the rater reliability and the convergent validity of the CSI-TH.

The CSI-TH was designed to be administered by a physical therapist, and its scoring scale spans from 0, indicating no significant association with cervical instability, to 16, indicating a strong connection to cervical instability. The 16 items included in the CSI-TH align with reported symptoms in patients with lumbar instability [48]. In the present study, among these 16 items, the one with the lowest occurrence rate, at 3.13%, was participants reporting feelings of neck instability, shaking, or a loss of control of the neck. Meanwhile, the highest occurrence rate, at 89.35%, was associated with participants expressing intolerance to extended periods of static posture (Figure 2). This observation is consistent with a study involving the lumbar instability questionnaire conducted by Chatprem et al., 2020 [48], which also recorded the highest percentage positive (100%) of participants reporting aggravated back pain symptoms when maintaining a static posture for an extended duration. This underscores the notion that pain exaggerates with prolonged posture, emerging as a predominant symptom in patients, even in cases involving instability in the neck or back area. 

For the rater reliability assessment, the CSI-TH was administered twice. Initially, inter-rater reliability between two physical therapists was assessed on the same day, based on the principle of evaluating differences between raters on the same occasion [48]. The result demonstrated excellent inter-rater reliability, reaching an ICC value of 0.983 (95% CI = 0.983–0.991). Subsequently, 48 h later, an assessment of intra-rater reliability was conducted. This short time-frame was chosen to avoid any potential clinical changes in the patient’s neck symptoms, with confirmation obtained by utilizing the GRC score, which indicated that participants had scores ranging between −2 and +3. These values remained within our accepted range of 3 [32]. The intra-rater reliability reached an ICC value of 0.995 (95% CI = 0.993–0.998). These reliability values confirm that physical therapists consistently achieved reliable results, whether one therapist used CSI-TH under various circumstances or different physical therapists used it at the same time.

The convergent validity showed a moderate correlation between the CSI-TH and the VAS-TH, mSBST-TH, and NDI-TH, with correlation coefficients (r values) of 0.5446, 0.5136, and 0.5545, respectively. These results were in line with our expectations, as these measurements differ in dimension and purpose. 

The VAS-TH was utilized to assess the intensity of pain sensation. The VAS-TH is effective in detecting pain, and there is robust evidence supporting its validity. A previous study found a moderate correlation between NDI and VAS scores (r = 0.691), which revealed that neck pain and disability depend on how each person with chronic neck pain perceives the impact of the pain on their ability to perform daily tasks [49]. Meanwhile, the current study showed a moderate correlation between the CSI-TH and the VAS-TH (r = 0.5446), which may be due to the different dominant measurement dimensions. While the CSI-TH encompasses the patient’s feelings, behaviors, activities, and adopted position, the VAS-TH solely focuses on pain intensity. However, the VAS-TH is commonly scored with a ruler, which is susceptible to bias or error [50]. 

The NDI-TH comprises six items related to limitations in daily-life activity and four items related to subjective symptomatology, including pain intensity, headache, concentration, and sleeping [51]. The moderate correlation observed between the CSI-TH and the NDI-TH (r = 0.5545) could be attributed to their shared dominance of neck disability and the level of symptoms, despite their differences in certain domains.

The mSBST-TH and the CSI-TH in this study exhibited a moderate correlation (r = 0.5136). While the mSBST-TH evaluates the risk of developing pain and explores subjective symptomatology and psychological factors during the experience of neck pain [43,44], the CSI-TH focuses on assessing the impact of pain on functional ability and disability without addressing psychosocial aspects. 

Among the 160 patients with nonspecific chronic neck pain, which may include some referred pain extending up to the head or down to the arms, there were 32 males and 128 females. Regarding participant characteristics, ages ranged from 35 to 55 years, with a mean age of 44.6 years. The number of females participants exceeded that of males, consistent with the proportion observed in other studies [34,52]

Globally, the number of incident cases of nonspecific neck pain was reported as 166.0 million (118.7 to 224.8) in females and 122.7 million (87.1 to 167.5) in males [53]. Previous studies have consistently indicated a higher likelihood of neck pain in women [54,55], possibly due to different biological factors, such as hormonal and neurobiological factors, as well as psychosocial factors, such as depression [56]. The complex relationship between sex and neck discomfort requires further clarification through sex-specific meta-analyses. 

Furthermore, the mean age of individuals found in previous studies to be suffering from neck pain aligns with the findings of the current study [21,34,41]. Most chronic pain is attributed to aging, which is a significant risk factor linked to development of cervical spondylosis and cervical instability.

The current study represents the first attempt to investigate the rater reliability of the CSI-TH and explore its relationship with another pain-related outcome. All items in the CSI-TH were revealed as having an adequate level of validity for identifying symptomatology, psychology, and functional disability in patients with cervical spine instability.

While the current study offers clinical utility, a few limitations warrant consideration. The absence of standardized methods to confirm participants’ cervical instability introduces a potential limitation. Additionally, our study did not exclude the possibility of neck pain originating from other sources, such as heart disease. Moreover, increasing the years of clinical experience among senior physiotherapists may enhance the confidence levels and applicability of our results.

## 5. Conclusions

The CSI-TH was designed for utilization by physical therapists and demonstrates robust reliability. It can be used by physical therapists, whether they are experienced or novices, and has an acceptable correlation to other neck-related questionnaires. The CSI-TH offers conciseness, is suitable for clinical applications, and is more budget-friendly compared to the gold standard in diagnosis for patients with cervical instability. However, the comparison between the CSI-TH and radiographic assessment still requires stronger clinical evidence. 

## Figures and Tables

**Figure 1 ijerph-20-06645-f001:**
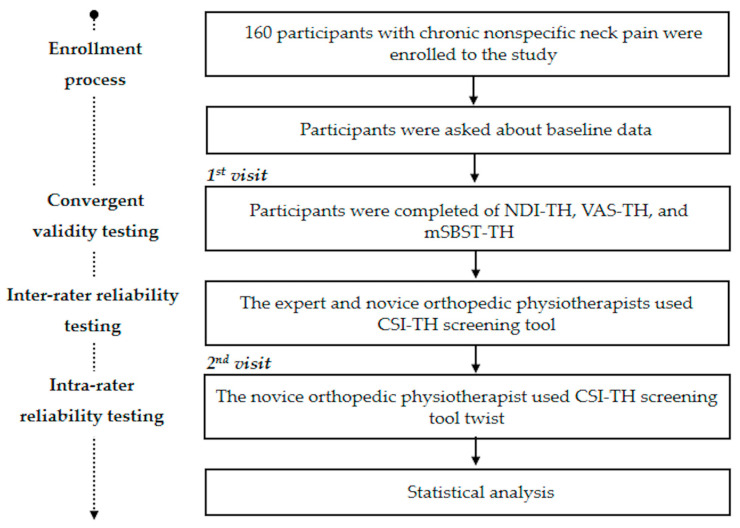
Overview of the study.

**Figure 2 ijerph-20-06645-f002:**
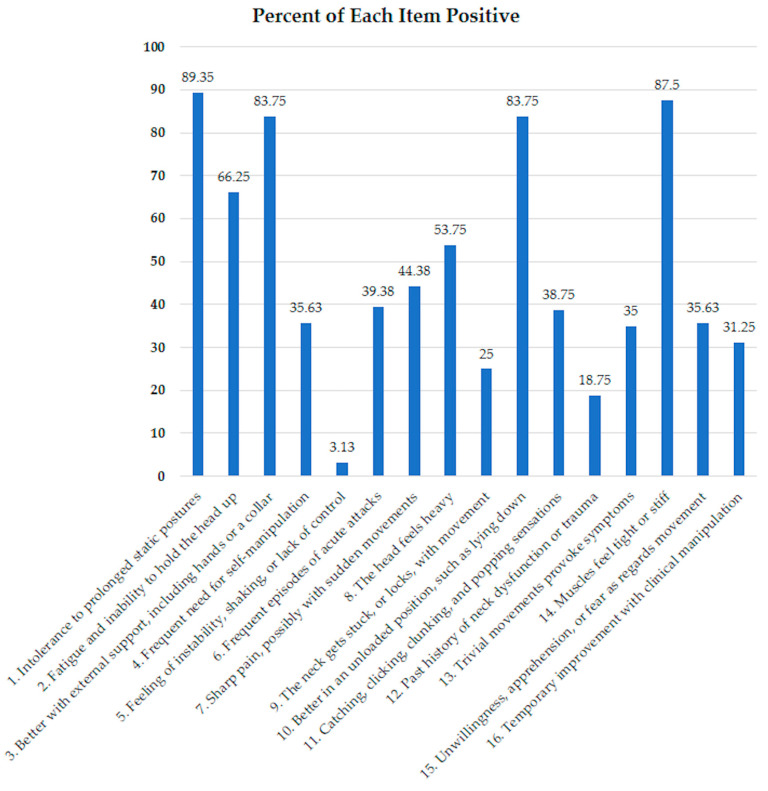
The percentage positive for each item of CSI-TH.

**Table 1 ijerph-20-06645-t001:** The cervical spine instability screening tool [21].

Items	Patients Respond	
	Yes	No
1. Intolerance to prolonged static postures		
2. Fatigue and inability to hold the head up		
3. Better with external support, including hands or a collar		
4. Frequent need for self-manipulation		
5. Feeling of instability, shaking, or lack of control		
6. Frequent episodes of acute attacks		
7. Sharp pain, possibly with sudden movements		
8. The head feels heavy		
9. The neck gets stuck, or locks, with movement		
10. Better in an unloaded position, such as lying down		
11. Catching, clicking, clunking, and popping sensations		
12. Past history of neck dysfunction or trauma		
13. Trivial movements provoke symptoms		
14. Muscles feel tight or stiff		
15. Unwillingness, apprehension, or fear as regards movement		
16. Temporary improvement with clinical manipulation		
**Total**		

**Table 2 ijerph-20-06645-t002:** Demographic characteristics of the 160 participants with nonspecific chronic neck pain.

	*n* (%)	Mean ± SD	Range
**Age (years)**		44.6 ± 6.5	35–55
**Gender**			
Male	32 (20.0)		
Female	128 (80.0)		
**Underlying disease**			
Yes	25 (15.63)		
No	135 (84.37)		
**Education level**			
Primary school	32 (20.0)		
High school	45 (28.13)		
University	83 (51.87)		
**Pain duration (months)**		15.48 ± 19.26	3 to 120
3–12	126 (78.75)		
>12	34 (21.25)		
**GRC score**		0.01 ± 0.45	−2 to 3

GRC: global rating of change.

**Table 3 ijerph-20-06645-t003:** The score of each questionnaire.

	Min	Q1	Median	Mean	Q3	Max
CSI-TH (score)	2	6	8	7.7	9	14
VAS-TH (score)	0.3	3.2	4.5	4.4	5.8	9.3
mSBST-TH (score)	0	3	4	4.3	5	9
NDI-TH (score)	0	6	11	10.9	16	30

CSI-TH: Cervical Spine Instability Thai version; VAS-TH: Visual Analog Scale Thai version; NDI-TH: Neck Disability Index Thai version; mSBST-TH: Modified STarT Back Screening Tool Thai version.

**Table 4 ijerph-20-06645-t004:** Reliability of CSI-TH screening tool among 160 participants with non-specific chronic neck pain.

	Mean ± SD		ICC	95% CI
	Expert	Novice		
Inter-rater reliability	7.70 ± 2.56	7.72 ± 2.60	0.987	0.983–0.991
Intra-rater reliability		7.68 ± 2.52	0.992	0.989–0.994

**Table 5 ijerph-20-06645-t005:** Correlation between the CSI-TH and the VAS-TH, NDI-TH, and mSBST-TH (*n* = 160).

Correlation	CSI-TH	VAS-TH	mSBST-TH	NDI-TH
CSI-TH	1.00	0.5446 *	0.5136 *	0.5545 *

* Statistically significant, *p* < 0.01; CSI-TH: Cervical Spine Instability Thai version; VAS-TH: Visual Analog Scale Thai version; NDI-TH: Neck Disability Index Thai version; mSBST-TH: Modified STarT Back Screening Tool Thai version.

## Data Availability

The data will be available for anyone who wishes to access them for research purposes, and contact should be made via the corresponding author: thiwch@kku.ac.th.

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
