# Peer review of "Cervical Spine Instability Screening Tool Thai Version: Assessment of Convergent Validity and Rater Reliability"

_ijerph, 2023, doi:10.3390/ijerph20176645_

Round 1
Reviewer 1 Report
The authors reported a assessment of the Cervical Spine Instability Screening Tool Thai Version in comparison with existing assessment. The methods and the study in general are well described. It is missing maybe a coefficient of variation computation but the results are clearly presented and could just be a bit improved with a COV report. correlation with coefficient between 0.5 and 0.6 are poor correlation coefficient.
The english is clear. Minor typo may be corrected.
Author Response
Thank you for your feedback. It could be used to improve my manuscript.
The change in your comment is highlighted in yellow.
Reviewer 2 Report
Dear Authors,
this is very interesting study. My only concern is about its local character.
You will find below some minor comments to improve your paper:
- some citations are missing or are far away from the place, where is should be, f.e. Rueangsri et al.;
- how did you verify if patients had nonspecific neck pain, not other disorders, simulating neck pain, like f.e. heart disease?
- the name and country of the company, that developed STATA software, is missing;
- Table 5 can be described in text;
- it would be worthful to include also quartiles in table 3 (for citing purposes, more data can be useful);
- all results should be included in the Results section, not in the Discussion (lines 257-260, fig. 2);
- there are some more limitations of the study like the experience of the "experienced physical therapist" (in my opinion it should be someone with min. 10 years of experience).
Author Response
Thank you for all your comments and suggestions. They hold great value for my work.
The changes that are related to your comment are reviewed by green highlight.
